# What Does Intracytoplasmic Sperm Injection Change in Embryonic Development? The Spermatozoon Contribution

**DOI:** 10.3390/jcm12020671

**Published:** 2023-01-14

**Authors:** Sandrine Chamayou, Filippo Giacone, Rossella Cannarella, Antonino Guglielmino

**Affiliations:** 1Centro HERA—Unità di Medicina della Riproduzione, Via Barriera del Bosco, 51/53, Sant’Agata li Battiati, 95030 Catania, Italy; 2Glickman Urological & Kidney Institute, Cleveland Clinic Foundation, Cleveland, OH 44195, USA; 3Department of Clinical and Experimental Medicine, University of Catania, 95123 Catania, Italy

**Keywords:** embryo development, embryo kinetics, epigenetic, ICSI, IMSI, male factor, male infertility, spermatozoa, sperm DNA fragmentation, time-lapse

## Abstract

The intracytoplasmic sperm injection (ICSI) technique was invented to solve severe male infertility due to altered sperm parameters. Nowadays, it is applied worldwide for the treatment of couple infertility. ICSI is performed with any available spermatozoon from surgery or ejaculated samples, whatever are the sperm motility, morphology or quantity. The aim of the present review was to study if embryo development and kinetics would be modified by (1) ICSI under the technical aspects, (2) the micro-injected spermatozoa in connection with male infertility. From published data, it can be seen that ICSI anticipates the zygote kinetics Furthermore, because fertilization rate is higher in ICSI compared to conventional in vitro fertilization (IVF), more blastocysts are obtained for clinical use in ICSI. Sperm and spermatozoa characteristics, such as sperm parameters, morphology and vitality, DNA content (levels of sperm DNA fragmentation, microdeletions, and chromosomal abnormalities), RNA content, epigenetics, and sperm recovery site (testicular, epididymis, and ejaculated), have an impact on fertilization and blastocyst rates and embryo kinetics in different ways. Even though ICSI is the most common solution to solve couples’ infertility, the causes of male infertility are crucial in building a competent spermatozoa that will contribute to normal embryonic development and healthy offspring.

## 1. Introduction

While in vitro fertilization (IVF) was mainly developed to solve gynecological infertility [1], the primary indications for intra cytoplasmic sperm injection (ICSI) were male-based due to altered semen parameters and total fertilization failure (TFF) [2,3]. Since the first clinical applications in 1992, the proportion of ICSI on conventional IVF did not stop to increase, getting stabilized at 70% of the fresh in vitro cycles since 2013 [4]. ICSI is the in vitro fertilization method of choice when using frozen-thawed oocytes [5] and surgically retrieved spermatozoa [6], as well as in case of preimplantation genetic testing (PGT) [7]. ICSI can be performed with frozen-thawed spermatozoa [8] and with immature spermatozoa [9]. Several variants exist, such as intracytoplasmic morphologically selected spermatozoa (IMSI), where spermatozoa are observed at 6600× (400× in conventional ICSI) and are thus selected based on organelle morphology [10], physiological intracytoplasmic sperm injection (PICSI) [11], and PIEZO-ICSI that is reported as being an effective insemination method in fragile oocytes [12].

In conventional IVF, the fertilization process occurs ‘naturally’, without human micromanipulation. The fertilizing spermatozoon goes through the zona pellucida (ZP) to fertilize the oocyte, as is the case in spontaneous conception. It is the opposite in ICSI, whereby one biologist-chosen spermatozoon is entirely micro-injected inside the ooplasm, bypassing the ZP binding/selection and gametes-fusion process.

By definition, ICSI ‘assists’ in vitro fertilization and allows the embryo formation (and offspring) in couples in which it would have been technically and/or biologically impossible, such as azoospermic patients due to the congenital bilateral absence of vas deferens (CBAVD) [13]; in patients with globozoospermia; or when using gametes with a reduced fertilizing power such as frozen-thawed oocyte [14]. By forcing the fertilization process, ICSI ignores the clinical/biological causes that would have prevented embryo formation. On technical and biological aspects, two questions arise: Does the ICSI technique have any consequences on embryonic development when compared to IVF? How does the micro-injected spermatozoon in ICSI modify the embryo development?

## 2. Comparing Fertilization Process between IVF and ICSI

Even if performed in vitro, the routine IVF mimics the natural fertilization process. The oocyte–cumulus–corona radiata complex is surrounded by motile spermatozoa that undergo acrosomal reactions releasing hyaluronidase enzymes to penetrate the cumulus–corona radiate complex and reach the ZP. Then, the spermatozoa head binds to the ZP3 glyco-protein and spermatozoa continue their way through the ZP to the perivitelline space. After that, one spermatozoon makes contact with the metaphase II oocyte membrane. The equatorial region of the sperm head touches the surrounding microvilli of the oocyte, and both gametes fuse together, creating continuity between the two plasma membranes. Immediately after, and as a result of gamete fusion, calcium ions in the submembrane vesicles of the oocyte are released and block the penetration of other sperm through the depolarization of the cytoplasmic membrane. In addition to preventing polyspermy, the release of Ca^2+^ stimulates respiration and the metabolism of the oocyte through an increase in intercellular pH and an increase in oxidative metabolism [15]. The spermatozoa nuclear chromatin that was very tightly packed starts to decondense after gametes fusion. The protamine proteins complexed with sperm DNA are replaced by histone proteins. The chromatin spreads out within the nucleus now called pronucleus and moves close to the nuclear material of the oocyte. The male DNA is demethylated whereas methylation is maintained in the female genome. In the meantime, the metaphase II oocyte completes the second meiosis division with the rapid succession of anaphase II and telophase II. The second polar body is released in the perivitelline space. Both pronuclei become visible [16]. DNA replication occurs in the developing pronuclei; one chromosome forms two chromatids as the pronuclei approach each other. Then, the pronuclei membranes break down and chromosomes intermingle. The one-cell stage embryo is now reached. Maternal and paternal chromosomes quickly organize around the mitotic spindle derived from the duplicated sperm centrosome.

In ICSI, the fertilizing spermatozoon penetrates neither the cumulus–corona radiata complex nor the ZP to fuse with the metaphase II oocyte as it is directly micro-injected into the ooplasm after immobilization. Immobilization is a necessary micromanipulation to make the sperm membrane permeable and allow the release of ‘sperm cytosolic factors’, which is necessary to oocyte activation [17]. After sperm micro-injection, the oocyte membrane depolarization due to calcium cortical vesicles calcium release is observable. The two pronuclei appear and fade at the zygote stage and the in vitro embryo development is similar in IVF-fertilized oocytes and ICSI-fertilized oocytes.

## 3. Does ICSI Modify Embryo Development? Comparison with Conventional IVF

In assisted reproductive technologies (ART), different key clinical and biological performance indicators (KPIs) are applied to assess the effectiveness of a procedure in a particular setting or in a specific group of patients [18,19]. Since ICSI assists with fertilization, the advantage of ICSI versus IVF is to increase the benchmark fertilization rate by 5% (80% in ICSI versus 75% in IVF) [19]. The superiority of ICSI in fertilization efficiency is highlighted in split insemination treatments, where sperm condition is suitable for both IVF and ICSI [20,21,22,23,24]. According to our data, the KPI ‘number of needed oocytes to produce one clinically usable blastocyst’ is inferior in ICSI compared to IVF. In other words, more blastocysts are produced from fewer oocytes in ICSI (2.3 metaphase II oocyte for 1 blastocyst) compared to IVF (2.9 metaphase II oocytes) [10]. This result is valid whatever the ovarian stimulation protocols, infertility indications or female age are.

Nowadays, the entire embryo development after in vitro fertilization is fully observable using incubators combined with time-lapse monitoring [25,26]. Morphokinetic parameters are used to identify the time of each cellular event from the second polar body extrusion right after sperm penetration to the fully hatched blastocyst looking for uterus contact and the duration of cell cycles [27].

In Bodri et al.’s study, a statistically significant anticipation of pronuclei fading (1.5 h) to 4-cell stage (1.1 h) were measured in ICSI embryos compared to IVF embryos [28].

This anticipation time observed at pronuclei fading decreases as the embryo development continues and disappears from the 5-cell stage to the blastocyst stage. The statistically significant anticipations of pronuclei fading, as well as 2-cell and 3-cell stages in ICSI embryos compared to IVF embryos, were confirmed by other groups using sibling oocytes [29] and oocytes from different cohorts [30,31].

It must be remembered that while the exact moment of spermatozoon penetration is known in ICSI, the fertilization process is longer in IVF and cannot be determined with exactness. In IVF, the sperm penetration through the cumulus cells and ZP, as well as gametes fusion, can be estimated to 1 hour duration [31]. In all studies, the cell cycles (cc2, cc3) and synchronizations of cell divisions (s2, s3) remained invariant according to the fertilization method. As a consequence, it can be concluded that ICSI anticipates the first cleavage stages because it shortens the two observable pronuclei durations but does not modify the embryo kinetics. If mathematical morphokinetic models are used for embryo selection, they must be adjusted at the cleavage stage depending on whether the embryos were ICSI- or IVF-derived [28].

The rates of euploidy, aneuploidy, and mosaicism are similar in sibling blastocysts derived from IVF or ICSI in couples with normozoospermia [32,33].

## 4. Spermatozoon Effect on Embryo Development

### 4.1. Sperm Parameters

ICSI is the chosen fertilization method when at least one sperm parameter is altered, such as sperm count (oligozoospermia), sperm motility (asthenozoospermia), or sperm morphology (teratozoospermia).

Many ICSI studies failed to correlate sperm parameter alteration after ejaculation with embryo development [34,35,36], concluding that ICSI resolves male infertility due to the semen parameters alteration. Nevertheless, data correlating the morphology of the micro-injected spermatozoon in ICSI or IMSI outcomes evidenced the paternal effect on embryo development. In ICSI, the micro-injection of morphologically abnormal spermatozoa can result in decreased fertilization and implantation rates [37]. In IMSI, the increasing volume of vacuoles in the micro-injected sperm head was correlated with a decreasing blastocyst rate [38,39,40] and a delay in embryo development following the pronuclear fading time and continuing all along the cleavage stage [40,41]. The sperm head vacuoles would be the sign of incorrect sperm DNA packaging [42], protamine organization [43], and DNA fragmentation [44].

Globozoospermia is a rare morphological sperm abnormality characterized by a round-headed spermatozoon due to a lack of acrosome and a coiled tail [45]. For those less than 0.1% of infertile men with globozoospermia, ICSI is the only solution to becoming a father with their own gametes. Compared to non-globozoospermic sperm, the fertilization rate is diminished. Oocyte activation fails due to sperm-specific phospholipase PLCζ under-expression or inactivity [46], defective sperm chromatin condensation, and sperm DNA damage [47].

In absolute asthenozoospermia, the fertilization rate is significantly decreased with regard to sperm vitality and origin [48]. The combination of oligoasthenoteratozoospermia was found to correlate with a delay of 4-cell and 5-cell stages and cell division synchronization s1 and s2 [49].

### 4.2. Sperm DNA

In the later stages of spermatogenesis, the spermatozoon undergoes molecular remodeling. Histone proteins are substituted by protamines and haploid sperm DNA is broken on one or two strands in several parts to occupy as little space as possible inside the sperm head. The histone–protamine transition occurs at the epididymis level. In addition to the physiological sperm DNA fragmentation (SDF), several clinical and environmental factors are known to have negative impacts on sperm DNA integrity, increasing the percentage of SDF. Numerous studies emphasize the direct relationship between SDF and male infertility and increased miscarriage rate after IVF or ICSI [50]. Whereas low SDF can be repaired by the competent oocyte, high levels of SDF were correlated with embryo morphokinetic delay observable from the time of of pronuclei fading to the morula stage [51,52,53,54]. The consequences are lower blastulation and pregnancy rates [55]. The adverse effects are stronger when both sperm DNA stands are broken [56].

Maternal age is known as the main factor of embryonic aneuploidy [57] but spermatozoon can contribute to aneuploidy too, especially in males with reduced sperm concentration [58,59]. Increased paternal age over 50 years old is associated with damaged DNA, lower blastocyst rate, and a significantly higher number of trisomic embryos [60]. Obviously, the risk of aneuploid embryos is particularly high in males with abnormal karyotype due to gonosomal aneuploidy (47,XXY; 47,XYY). The percentage of euploid embryos to transfer after preimplantation genetic testing (PGT) in ICSI couples in which one member carries an abnormal karyotype (PGT-SR) varies according to the carrier partner (male or female) and the chromosomal abnormality [57]. When the structural chromosomal abnormality is carried by the male patient, the percentage of transferrable embryos is higher in Robertsonian translocation instead of reciprocal translocation. This result highlights that specific (structural) chromosomal abnormalities carried by the male partner are incompatible with embryo development and induce embryo arrest [57].

The Y chromosome is specific to males and can be microdeleted in Yq11 at the specific loci AZFa, AZFb, and AZFc, causing genetic infertility [61]. Minimal data correlating the effect of AZF microdeletion with embryo development and kinetics are available. Fertilization rate was found to be lower with sperm from AZF-microdeleted patients, but embryo development was similar [59].

In azoospermic patients with CBAVD due to cystic fibrosis mutation(s) and carrier couples that are candidates for PGT for a specific genetic disease, no study evidenced a correlation between the genetic trait and embryo development and kinetics.

### 4.3. Sperm RNA

Several molecular biology techniques, particularly RNA sequencing, have allowed the characterization of the whole RNA content of the sperm cell, which, as in other mammals, includes both coding and non-coding RNAs, such as mRNA, miRNA rRNA, piRNAs, lncRNA, siRNA, tRFs, and others [62,63,64,65]. Innovatively, some RNAs are encoded by genes packaged within the H2M4me3 histones, which are compatible with transcription. Hence, it has been recently speculated that the spermatozoon might be able of de novo transcription, at least in specific DNA loci [66]. A number of transcripts encoding for growth factors, transcription factors, or protein kinases have been identified in human sperm to a different extent in terms of concentration in infertile patients compared to fertile controls [67]. Some sperm RNAs could be effectors of male infertility by their injection in the oocytes [68,69,70]. As such, these RNAs, including clusterin, calmegin, and the integrator complex subunit I mRNA, seem to be defective in unfertilized oocytes and would play a role in early embryogenesis [70,71]. From mice studies, non-coding RNAs acquired during the epididymal transit play a role in embryogenesis [72]. Regarding the embryo kinetics, the miRNA appears to delay the cleavage stage from 2-cell to 5-cell stages and decrease the percentage of high-quality embryos [73].

A pilot study analyzing levels of several sperm-carried mRNAs encoding for genes involved in fertilization events, oocyte activation, chromatin remodeling, and DNA repair in oligoozoospermic patients and normozoospermic controls reported significantly lower levels of 21 mRNAs (e.g., mRNA of AKAP4, PTK7 PLCζ, POU5F1) in oligoozoospermic patients. A total of 90% of the degenerated embryos did not reach the morula stage in those patients [74]. Furthermore, a study on normozoospermic males undergoing ICSI with young donor oocytes found 324 small RNAs (including 5′-tRF-Asp-GTC; 5′-tRF-Phe-GAA, let-7f-2-5p, miR-4755-3p, miR-92a-3p, etc.) differently expressed according to cases with high or low blastocyst rates [75].

Emerging evidence has addressed to sperm RNA a role in early embryo development and embryo kinetic [76].

### 4.4. Sperm Epigenetic

Epigenetics involves mitotically and/or meiotically inheritable changes in gene function without alterations in DNA sequences, enabling the transformation of the same genome into several different transcriptomes. Spermatozoa have a unique epigenetic signature with a specific methylation profile [77]. In the fraction of the sperm genome that does not undergo the histone–protamine transition, retained histones are subjected to chemical modifications, such as methylation, acetylation, phosphorylation, and ubiquitination [78,79], that regulate genome activation and silencing [80]. Data suggest that these epigenetic factors may affect transcriptional regulation during embryogenesis and contribute extra-genomically to early embryonic development. Moreover, alterations in this highly specialized chromatin architecture may be associated with male infertility (decreased sperm concentration, motility, and fertilization ability) and embryo developmental anomalies [81,82].

### 4.5. Source of Spermatozoa

The spermatozoa for micro-injecting in ICSI can be recovered from ejaculated specimen or chirurgical extractions (e.g., testicular sperm extraction TESE, testicular sperm aspiration TESA, microsurgical epididymal sperm aspiration MESA, percutaneous epididymal sperm aspiration PESA) in case of azoospermic men [83]. According to sperm origin, the fertilization rate is higher with ejaculated sperm, followed by epididymal, and then testicular samples [6]. Different studies underlined the effect of sperm recovery on embryo kinetics comparing data from ejaculated semen with spermatozoon from testicular or epididymis. The micro-injection of spermatozoon recovered from the testicle (TESE, TESA) has an effect at the zygote stage with an earlier second polar body extrusion, a delay of pronuclei appearance, and a longer pronuclear stage [84,85,86]. At the cleavage stage, the embryos reach the 3-, 5-, 7-, 8-, and 9+ cell stages earlier in the case of testicular spermatozoon [85,86]. The morula and blastocyst stages are reached later [84,85,86]. A higher percentage of unequal cleavage from the 1-cell stage to the 3-cell stage are observed [84].

In the case of epididymal spermatozoon, embryo kinetics seem more similar to ejaculated semen, except for when they reach the 2-, 4-, and 6-cell stages. The blastocyst stage is delayed compared to testicular and ejaculated semen [85]. More blastocysts are obtained with ejaculated and normospermia compared to surgically extracted spermatozoon [85].

The effects of sperm recovery on embryo kinetics highlight, once again, the molecular aspects of spermatozoa maturation along the male genital tract. At the testicular level, the sperm DNA is enfolded with histones, is not yet compacted, and is less fragmented. The variation of the zygote kinetics for testicular spermatozoa compared to epididymis or ejaculated sperm is explained by the fact that the oocyte does not need to substitute protamines with histones or to repair fragmented DNA sperm as it would have done with a more mature sperm. The frequent unequal cleavage from the 1-cell stage to the 3-cell stage observed with testicular spermatozoa is explained by an incomplete maturation of centriole maturation in the testicular spermatozoon; this maturation being completed once the cell reaches epididymis [84]. In addition to the role of motility gain as the spermatozoon runs along the epididymis, the epididymis plays a role in molecular sperm maturation [87,88]. The incomplete molecular maturation of epididymis spermatozoa would explain the poor embryo development and lower ICSI outcomes compared to ejaculated sperm [76,89].

### 4.6. Sperm Cryopreservation

Even if semen cryopreservation decreases the levels of sperm mRNA [90,91], embryos from cryopreserved sperm have comparable development and kinetics compared to embryos from freshly ejaculated sperm [92,93]. The adverse effects of sperm cryopreservation are the loss of sperm motility and viability [94], but the sperm vitrification protocol is epigenetically safe and induces minor biological changes compared to conventional freezing [95].

The cryopreservation of testicular or epididymal sperm in patients with obstructive or non-obstructive azoospermia were found to have no impact on ICSI outcomes and embryo development compared to fresh testicular or epididymal samples [96,97].

### 4.7. Other Sperm Causes

According to scientific literature, other sperm causes can affect embryo development and kinetics. The high levels of reactive oxygen species in semen were associated with lower embryo quality and lower blastocyst rate [98]. The abundance of sperm proteins, such as those of the chaperonin-containing T-complex, correlate with early embryo quality and could be considered a predictive biomarker of ICSI outcomes in couples with idiopathic infertility [99].

## 5. Discussion

ICSI has been a revolution in the treatment of couple infertility due to severe male factor. Since 1992, the efficacy of ICSI prompted its use in all IVF laboratories around the world [4]. Due to the technical protocol, ICSI provides access to gametes observation following the micro-injection and the possibility of correlating cell quality/morphology with ICSI outcomes prospectively [16]. Thanks to the last generation of incubators equipped with time-lapse technology, it appears that the invasiveness of ICSI has no adverse consequences on embryo development. In contrast, it increases in vitro treatment, leading to an increased number of zygotes and a consequently higher number of blastocysts to transfer or freeze per cycle [19,24].

Nevertheless, if we go deeper into the observation of in vitro embryo development, according to spermatozoa morphology, sperm DNA configuration, sperm RNA content, or sperm source, we observe a direct effect of spermatozoa maturation on embryo development, pregnancy chances, and miscarriage risks. The studies that correlate with embryo kinetics and ICSI outcomes highlight the importance of the sperm maturation process through the male genital tract from the testicle to the epididymis and ejaculation. A fertilizing spermatozoon is not only 23 mono-chromatin chromosomes and a centriole, but it contains specific coding and non-coding RNA molecules that have an emerging role after the early embryo stages and an epigenetic signature, whereby any disruption can lead to consequences for the future embryo and offspring.

From the present review, it can be seen that fertilization rate is impacted by spermatozoa morphology (terato- and specifically globozoospermia), spermatozoa vitality (absolute asthenozoospermia), sperm DNA microdeletion (AZF), sperm RNA content, sperm epigenetics, and spermatozoa source (testicular, epididymis, and ejaculated). The zygote kinetics varies according to the morphology of the micro-injected sperm and, in particular, to the presence of head vacuoles. This result may be explained by a possible high level of SDF. In the same way, embryo kinetics from the cleavage to the morula stage and blastocyst rate are affected by sperm head vacuoles, high levels of SDF, and epigenetic factors. The variations in fertilization rate and zygote–embryo–blastocyst kinetics in the function of the sperm recovery site (testicular, epididymis, and semen) confirm how much the structural state and sperm molecular content impact embryo development after the first zygote moments (see Table 1). Interestingly, sperm cryopreservation has no consequences on embryo development.

Of course, our knowledge on sperm causes and effects on embryo development remains limited because we do not cover the precise state of DNA fragmentation and the coding/non-coding RNA content of the micro-injected spermatozoon. Whereas in natural fertilization the female reproductive tract strictly selects the sperm subpopulation that is allowed to approach the oocyte to fertilize through molecular and genetic compatibility processes [100], in the IVF lab, the biologist is trained in micromanipulation and chooses the sperm to micro-inject based on the best mobility and morphology. Actually, different strategies are under development to select the best/perfect e spermatozoa. These strategies are non-invasive for the spermatozoa and want to mimic the female reproductive tract selection [101]. They are based on polarized-light-selected sperm with birefringent heads; on sperm membrane characteristics eliminating apoptotic cells through magnetic microspheres–annexin V columns, the outer sperm membrane configuration and thus the morphology-incubating sperm in media–hyaluronic acid plates and sperm motility make the cells pass through microfluid channels. However, the lack of data on the efficacy of these techniques and the surplus costs for sperm preparation compared to the classical sperm preparations based on sperm washing, swim-up, or density gradient centrifugation detract from their wide application in ART laboratories.

Until recently, the causes for unsolvable fertility were mostly considered to be female-centered, such as advanced maternal age. However, the spermatozoon contribution to parenthood must now start to be considered. Advanced paternal age correlates with increased de novo mutations [102]. Very little is still known about the possible amplification effect of AZF microdeletions [60]. External factors, such as diet, environment, and oxidative stress, appear to compromise gametes functionality of the oocyte, as well as the spermatozoa, and consequently have adverse effects on offspring health [98,103]. To date, more than 2 million babies were born with ART, mainly with ICSI [104]. Compared to natural conceptions, the risk of major chromosomal defects in babies born from ART and especially from ICSI increases by 42% in case the of male infertility [105]. Imprinting disorders in the children have been associated with ICSI and paternal contribution [106,107].

## 6. Conclusions

As with the oocyte, the spermatozoon contributes to a healthy embryo development. Even if ICSI is the most common solution to resolve infertility in couples, we must not forget the causes of male infertility and the possible consequences on spermatozoon competence. The future development of non-invasive diagnostic instruments of the spermatozoon in order to know its biochemical structure and DNA/RNA content before microinjection would be determinant on the understanding of spermatogenesis, as well as the consequences of male infertility causes and an unfavorable environment on the building of a competent spermatozoon. In a world in which sperm quality has not stopped declining through the decades and in different continents due to adverse environments [108], a responsible and global commitment to resolving male infertility issues is the only way to improve male fertility and resolve couple fertility without ART.

## Figures and Tables

**Table 1 jcm-12-00671-t001:** Bibliography referencing the sperm/spermatozoon characteristics that modify embryo development.

Sperm/Spermatozoon Characteristic	Fertilization Rate	Blastocyst Rate	Zygote Kinetic	Cleavage Kinetic	Morula Kinetic	Blastocyst Kinetic
Oligoastenoteratozoospermia				[49]		
Absolute astenozoospermia	[48]					
Globozoospermia	[46]					
Abnormal spermatozoon morphology	[37]					
Sperm head vacuole		[38,39,40]	[40,41]	[40,41]		
Increased SDF		[51,52,53,54]	[51,52,53,54]	[51,52,53,54]	[51,52,53,54]	
Chromosome abnormality						[57,60]
AZF microdeletions	[59]					
RNA content		[73,74,75]		[73]		
Epigenetic	[81,82]	[81,82]				
Sperm recovery site						
Testicular	[6]		[84,85,86]	[84,85,86]	[84,85,86]	[84,85,86]
Epidydimis	[6]	[85]		[85]		

SDF: sperm DNA fragmentation.

## Data Availability

Not applicable.

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
