# Peer review of "What Does Intracytoplasmic Sperm Injection Change in Embryonic Development? The Spermatozoon Contribution"

_jcm, 2023, doi:10.3390/jcm12020671_

Round 1

Reviewer 1 Report

 I found that this paper to be overall very well informed, objective and well written, however certain points need to be clarified before the manuscript could be considered for publication.

·        The abbreviations (ICSI) should be avoided in the title.

·        All abbreviations in the abstract should be written out first time use except famous abbreviations (RNA & DNA).

·        Comparing fertilization process between IVF and ICSI should be provided in Table.

·        Does ICSI modify embryo development? Comparison with conventional IVF should be provided in Table.

·        The authors should provide Figure clarify the spermatozoon effect on embryo development.

·        Please, provide future perspective or recommendation of further studies.

Author Response

Reviewer #1:

The abbreviations (ICSI) should be avoided in the title: Answer: ICSI was substituted by signification in the title.

All abbreviations in the abstract should be written out first time use except famous abbreviations (RNA & DNA). Answer: ICSI and IVF were substituted by signification in the abstract.

Comparing fertilization process between IVF and ICSI should be provided in Table. Does ICSI modify embryo development? Comparison with conventional IVF should be provided in Table. The authors should provide Figure clarify the spermatozoon effect on embryo development. Answer: We agree that an illustration (table or figure) would be didactic to show the difference in fertilization process between IVF and ICSI. Nevertheless, the full comparison between IVF and ICSI process is not the subject of the present manuscript. To support this argument, we underline that in the present manuscript we compared neither IVF and ICSI morphokinetic parameters nor IVF/ICSI efficiency as it was done elsewhere (see our article Chamayou et al. 2021 – reference 24). The subject we chose was how the spermatozoon contribution due to male fertility cause can influence embryo development but not the micro-injection technique by itself (see paragraph 2).

Please, provide future perspective or recommendation of further studies. Answer: the following part was added in the conclusion paragraph: ‘The future development of non-invasive diagnostic instruments of the spermatozoon in order to know its biochemical structure and DNA/RNA content before microinjection would be determinant on the understanding of spermatogenesis and the consequences of male infertility causes and an unfavorable environment on the building of a competent spermatozoon.’.

Reviewer 2 Report

Lines 33-34: "While in vitro Fertilization (IVF) has been developed to solve gynecological infertility [1],".

Although, this notion is common, it is not generally accepted. Take into consideration that from the very first days of IVF, cases with male factor infertility were treated, though with low success.

Line 46: "In conventional IVF, the fertilization process occurs ‘naturally-like’".

The notion that conventional IVF is a "natural" method of fertilization is misleading. It is better to say that it is less invasive than ICSI.

Line 167: please, replace "donor oocyte" with "normal oocyte" or "competent oocyte"

Line 206: "RNAs acquired during the epididymal transit plays"

please use "play" instead of "plays" (RNAs is in plural)

Line 289: "micro-injection" instead "micr-injection"

Lines 324-330: Here, it would be nice to suggest the good training, of biologists performing ICI, on sperm morphology. Unfortunately, in most IVF units the training of biologists  is almost exclusively focused on micromanipulations and not on the characteristics of sperm morphology

Line 347: "it must not make forget"

Better write: "it must not make us to forget"

Author Response

Lines 33-34: "While in vitro Fertilization (IVF) has been developed to solve gynecological infertility [1],".

Although, this notion is common, it is not generally accepted. Take into consideration that from the very first days of IVF, cases with male factor infertility were treated, though with low success. Answer: the word ‘mainly’ was introduced in the sentence to contrast the infertility indication:  IVF was mainly developed to solve gynecological infertility…

Line 46: "In conventional IVF, the fertilization process occurs ‘naturally-like’". The notion that conventional IVF is a "natural" method of fertilization is misleading. It is better to say that it is less invasive than ICSI. Answer: the sentence was modified as follows ‘In conventional IVF, the fertilization process occurs ‘naturally’ because without human micromanipulation.’.

Line 167: please, replace "donor oocyte" with "normal oocyte" or "competent oocyte": Answer: modification was made.

Line 206: "RNAs acquired during the epididymal transit plays" please use "play" instead of "plays" (RNAs is in plural): Answer: modification was made.

Line 289: "micro-injection" instead "micr-injection" Answer: modification was made.

Lines 324-330: Here, it would be nice to suggest the good training, of biologists performing ICI, on sperm morphology. Unfortunately, in most IVF units the training of biologists  is almost exclusively focused on micromanipulations and not on the characteristics of sperm morphology. Answer: we added ‘is trained to micromanipulation and’

Line 347: "it must not make forget"

Better write: "it must not make us to forget" Answer: modification was made.

Round 2

Reviewer 1 Report

The revised manuscript is greatly improved.  I recommend the paper for publication.